# Study on the Moving Target Tracking Based on Vision DSP

**DOI:** 10.3390/s20226494

**Published:** 2020-11-13

**Authors:** Xuan Gong, Zichun Le, Hui Wang, Yukun Wu

**Affiliations:** 1College of Computer Science and Technology, Zhejiang University of Technology, Hangzhou 310023, China; 1111712011@zjut.edu.cn (X.G.); 1111712012@zjut.edu.cn (H.W.); 1121712004@zjut.edu.cn (Y.W.); 2College of Science, Zhejiang University of Technology, Hangzhou 310023, China

**Keywords:** KCF, DSST, vision DSP, SIMD, iDMA, runtime, data parallelism, instruction parallelism

## Abstract

The embedded visual tracking system has higher requirements for real-time performance and system resources, and this is a challenge for visual tracking systems with available hardware resources. The major focus of this study is evaluating the results of hardware optimization methods. These optimization techniques provide efficient utilization based on limited hardware resources. This paper also uses a pragmatic approach to investigate the real-time performance effect by implementing and optimizing a kernel correlation filter (KCF) tracking algorithm based on a vision digital signal processor (vision DSP). We examine and analyze the impact factors of the tracking system, which include DP (data parallelism), IP (instruction parallelism), and the characteristics of parallel processing of the DSP core and iDMA (integrated direct memory access). Moreover, we utilize a time-sharing strategy to increase the system runtime speed. These research results are also applicable to other machine vision algorithms. In addition, we introduced a scale filter to overcome the disadvantages of KCF for scale transformation. The experimental results demonstrate that the use of system resources and real-time tracking speed also satisfies the expected requirements, and the tracking algorithm with a scale filter can realize almost the same accuracy as the DSST (discriminative scale space tracking) algorithm under a vision DSP environment.

## 1. Introduction

Moving target tracking is a process of processing and analyzing the video (sequence) images captured by photoelectric sensors and making full use of the information collected by sensors to track and locate the target. Target tracking is also a basic function of computer vision applications and is widely used in the fields of intelligent monitoring [1], pose estimation [2], motion recognition [3], behavioral analysis [4], and automatic driving, among others. In recent years, vision tracking based on correlation filters [5,6,7,8,9,10], such as kernel correlation filters (KCFs) [5,6] and correlation filters with scale estimation (DSST) [7,8], has become a research hotspot due to their advantages in terms of speed and efficiency.

The correlation filter algorithm trains the filter by regressing the input features to the target Gaussian distribution and locates the target position by searching for the response peak in the predicted distribution during follow-up tracking. The fast Fourier transform (FFT) is skillfully applied to realize a large speed increase. Many extended methods that are based on correlation filters are available, which include kernel correlation filters and correlation filters with scale estimation.

In recent years, research on correlation filters in target tracking has made substantial progress. Bolme et al. [9] applied the designed minimum output sum of the squared error (MOSSE) adaptive correlation filter to target tracking. The algorithm only requires a sample image of the target region for training the target model, and the similarity calculation between the target and all candidate regions is conducted using the discrete Fourier transform (DFT), which substantially increases the algorithm speed. Henriques JF et al. [5] proposed the circulant structure of tracking by detection with kernels (CSK). The algorithm shifts the candidate image blocks circularly and obtains many samples for the training classifier. The training and detection of these classifiers are based on the dot product operation in the frequency domain, which substantially increases the tracking speed. Later, Henriques JF [9] improved the CSK algorithm. Multidimensional feature tracking was conducted by extracting the image histogram of oriented gradients (HOG) features [11]. Kernel correlation filter and dual-core correlation filter (DCF) tracking algorithms were proposed to further improve the tracking performance. Due to its satisfactory robustness regarding illumination and high tracking speed, the KCF algorithm is more suitable for embedded systems.

Many scholars that investigate the KCF algorithm focus mainly on the traditional target tracking problems, such as scale changes and severe occlusion [12,13]. However, in practical scenarios, the realization of real-time performance of tracking algorithms remains challenging, especially in the embedded field.

In recent years, with the rapid development of high-speed DSPs (digital signal processors), various high-performance hardware architectures can satisfy the real-time requirements of data processing (e.g., Cadence Inc.’s vision DSP). Researchers have conducted studies based on hardware platforms. Their real-time image tracking systems were based mainly on three types of hardware architectures: FPAG (Field Programmable Gate Array) and DSP architectures, multi-core DSP architectures, and single-core DSP architectures [14,15,16,17,18,19,20,21]. Most of these embedded system studies of visual tracking are based on traditional hardware architectures. For example, in [22], the author utilizes EDMA (enhanced direct memory access) and the caching technique to optimize the Camshift algorithm, which is based on the DSP of TI (Texas Instruments) company, and a frame rate of 3 fps can be realized. Liu W [23] optimizes the tracking algorithm from a programming perspective based on the DSP of TI company by including an option for setting the compiler optimization and improving the code loop structure, among other changes. Finally, a frame rate 25 fps can be realized. Shi X D et al. [24] optimize the KCF algorithm based on DSP from a software perspective in terms of the functionality and memory allocation, among other aspects, and after optimization, a frame rate of 25 fps can be realized.

In practical application, embedded devices often need high real-time performance. Therefore, whether it is a traditional embedded chip (ARM, DSP) or a powerful artificial intelligence chip (vision DSP), system level optimization based on resource-limited environments is essential, especially for computing vision applications with high real-time requirements, such as target detection, target tracking and so on. In the paper, our objective is to investigate the impact factor of the embedded system on the tracking algorithm. The performance of the algorithm, such as tracking precision, is not the focus of this research phase, so we do not further improve the KCF tracking algorithm and extensive performance evaluation. 

Contributions of this paper are as follows: (1) We use a pragmatic approach to investigate the real-time performance effect factors in the embedded system by implementing and optimizing KCF based on a vision digital signal processor (vision DSP), including evaluating and analyzing the impacts of DP (data parallelism), IP (instruction parallelism), and the characteristics of parallel processing of the DSP core and iDMA (integrated direct memory access) in the tracking system; (2) We propose a time-sharing strategy to resolve storage issues of limited DSP local memory for tracking data. As an additional contribution, we introduced a scale filter to overcome defects of KCF for scale transformation. These research results are also applicable to other machine vision algorithms and other embedded systems with similar structures. Experiments show that our method can effectively improve the real-time performance of the tracking algorithm under ensuring the performance of the algorithm. 

In the second section of this paper, we introduce the vision DSP hardware platform from Cadence Company. In the third section, the paper introduces the KCF algorithm and applies a scale filter to overcome the scale change limitation of the KCF algorithm. In the fourth section, we optimize the entire algorithm structure and data processing from a system optimization perspective. In the fifth section, an experiment compares the algorithm performance and analyzes system resource usage. The sixth section summarizes the research results.

## 2. Hardware Platform

The DSP for processing images has common features of the hardware structure and instruction set. Cadence Company’s vision DSP (Cadence, San Jose, California, USA) is a high-performance processor that supports SIMD-VLIW (single instruction multiple data, very long instruction word). Each instruction has a maximum of 64-way SIMDs and one to five issue slots, which supports 8/16/32-bit integer vector operations and 16/32-bit half-precision and single-precision vector floating point units (VFPUs). Two 128-bit AXI interfaces are used for system memory and peripherals: one AXI is used for DSP load/store instructions and fetch operations, and the other AXI interface is used for iDMA to move data between DDR and DRAM. In storage, the DSP includes two 256 K DRAMs (DRAM0 and DRAM1) and a 32 K RAM and vector register array. In addition, the DSP maximum working clock frequency is 600 MHz. The hardware structure is illustrated in Figure 1.

As a high-performance vision processor, DP (data parallelism) and IP (instruction parallelism) are the first prerequisites. This paper utilizes methods to realize DP and IP to improve the performance of the algorithm. DP mainly refers to SIMD (single instruction multiple data). In vision DSP, the width of the SIMD is also the vector width. Figure 2 compares scalar and SIMD for multiplications. Scalar operations compute a single data element at a time, whereas SIMD operations replicate an operation for multiple data elements in a vector.

Vision DSP uses the vector register as the calculation unit, and the vector size is 512 bits. The number of vector elements depends on the element width. If the element width is 8 bits, the number of vector elements is 64, as presented in Figure 3.

Instruction parallelism refers mainly to VLIW (in vision DSP, the instruction width of VLIW can be 64 or 128 bits). Usually, VLIW instruction can perform multiple operations in the same cycle. The number of operations that are executed in parallel depends on the instruction format. In vision DSP, there are five instruction issue slots, and different instruction formats occupy different sets of issue slots. Figure 4 presents five instruction issue slots and specifies the types of operations that are performed by each slot, e.g., ALU(Arithmetic Logic Unit) operations can be issued in four slots, whereas multiply and select operations can be issued in only one slot separately. The number of operations the vision DSP can issue is flexible; typically, between 1 and 5 operations can execute in parallel.

Vision DSP also provides rich vector register types and related operations. At the same time, Cadence provides a powerful simulation environment, namely, Xplorer, which not only functions as the integrated development environment (IDE) in the general software but also can profile the system resources at runtime and generate the disassembly code for further investigation of the instructions. The study in this paper is based mainly on the soft simulation environment.

## 3. Kernel Correlation Filter

In this section, we will provide an introduction of the KCF tracking algorithm and on how the scale filter is to be introduced to the kernel correlation filter.

### 3.1. Algorithm Introduction

The KCF is a discriminative tracking algorithm that is divided into two stages: sample training and target detection. In the sample training stage, the target is selected and is shifted circularly to obtain positive and negative samples (the selected base sample is taken as a positive sample, and the virtual sample after cyclic shift for the base sample is taken as a negative sample). Ridge regression is used to train these samples to obtain the target classifier. In the detection stage, the correlation coefficients of the selected region and the tracking target are calculated by the kernel function, and the selected region with the largest correlation coefficient is selected as the new target. The algorithm uses a two-dimensional fast Fourier transform (FFT2D) to map the time domain to the frequency domain, which reduces the amount of computation in the processes of sample training and target detection. Finally, in the process of fast target detection, the two-dimensional inverse fast Fourier transform (IFFT2D) is used to map the frequency domain to the time domain to identify the optimal target location in the next frame. Next, we will introduce the target tracking process of the KCF algorithm from three aspects: training sample set, target classifier and detecting new target.

#### 3.1.1. Training Sample Set 

The training sample set required by the KCF tracking algorithm is generated by a cyclic shift of samples in the target region. During the sample training, the selected target area in the first frame is taken as the base sample x. Then, cyclic matrix P is used to carry out a cyclic shift of the base sample to obtain the training sample set $\left\{P^{i}x|i=0,\ldots ,n-1\right\}$.
(1)$$P=\left[\begin{array}{ccccc} 0 & 0 & \ldots & 0 & 1\\ 1 & 0 & \ldots & 0 & 0\\ 0 & 1 & \ldots & 0 & 0\\ \vdots & \vdots & \ddots & \vdots & \vdots \\ 0 & 0 & \ldots & 1 & 0 \end{array}\right]$$

If $X_{i}$ is the training sample after the base sample moves i positions circularly, then data metric X can be generated by $X_{i}=P^{i}x$, i= 0,…n−1.
(2)$$X=C\left(x\right)=\left[\begin{array}{ccccc} x_{1} & x_{2} & \ldots & x_{n-1} & x_{n}\\ x_{n} & x_{1} & \ldots & x_{n-2} & x_{n-1}\\ x_{n-1} & x_{n} & \ldots & x_{n-3} & x_{n-2}\\ \vdots & \vdots & \ddots & \vdots & \vdots \\ x_{2} & x_{3} & \ldots & x_{n} & x_{1} \end{array}\right]$$

Equation (2) can be diagonalized by discrete Fourier transform (DFT), Equation (3): (3)$$X=F\mathrm{diag}(\overset{\wedge }{x})F^{H}$$

F is a constant matrix independent of the base sample, $F^{H}$ is conjugate transpose of F and $\overset{\wedge }{x}$ denotes the DFT of x,$\overset{\wedge }{x}=$
ℱ(x).

#### 3.1.2. Train Sample Classifie

The KCF tracking algorithm introduces the kernel function f(z) on the basis of the ridge regression classifier to improve computing performance by mapping low dimension vector x to high dimension, and the optimal solution of the classifier that is trained by the minimal regularization risk function can be expressed as:(4)$$\sum_{w}^{\min}{\left(<w,\;\varphi \left(\chi_{i}\right)>-y_{i}\right)}^{2}+\lambda {\left\|w\right\|}^{2}$$
where $y_{i}$ represents the desired output, λ represents a regularization parameter that is used to control overfitting, $x_{i}$ represents the training sample, φ(x) represents a mapping function for the kernel space, w represents a linear optimal solution for a combination of samples that is obtained via Equation (4)
(5)$$w=\sum_{i}\alpha_{i}\varphi (x_{i})$$

The intention of Equation (4) is to find a classifier that satisfies the condition. The classifier f(z) can be denoted via Equation (5):(6)$$f(z)=w^{T}z=\sum_{i}a_{i}k(z,x_{i})$$

$k(x_{i},x_{j})$ is the kernel function. According to reference [25], vector α can be denoted as:(7)$$\alpha \;=\;(K+\lambda I{)}^{-1}y$$

The vector α is composed of sample training parameters, I is the identity matrix, K is kernel matrix and generated by the kernel function:(8)$$K_{i,j}=k(x_{i},x_{j})$$

The DFT result of Equation (7) using Equation (3) can be denoted as following:(9)$$\hat{\alpha }=\frac{\hat{y}}{{\hat{k}}^{\mathrm{xx}}+\lambda }$$

$k^{xx}$ is the first line element of kernel matrix K.

#### 3.1.3. Target Detection

The regression function of the classifier with new image block z can be denoted as below:(10)$$f(z)=w^{T}z=\sum_{i}a_{i}k(x_{i},z)$$

The candidate image block of z can be obtained by cyclic matrix. $K^{z}$ is defined as the kernel matrix of the training sample and all the candidate image blocks. According to the unitary invariant kernel function theorem, it can be known that $K^{z}$ is the cyclic matrix:(11)$$K^{z}=C\left(k^{xz}\right)$$

Here, $k^{xz}$ represents the kernel correlation operation between x and z. According to Equations (10) and (11), the response functions of all candidate image blocks can be obtained:(12)$$f(z)={\left(K^{z}\right)}^{T}\alpha$$

The objective of tracking is to calculate the location of the maximum response value of the new block z. The result of DFT for Equation (12) is as below:(13)$$\overset{\wedge }{f}(z)=\overset{\wedge }{k^{xz}}⊗\overset{\wedge }{\alpha }$$

The symbol “⊗” denotes the dot product. The correlation coefficient responses of all candidate image blocks are obtained via the IDFT transform, and the position with the largest correlation coefficients is the best position for the target to appear in the frame.

### 3.2. Applying a Scale Filter TO Kernel Correlation Filter

Equation (13) in the previous section can also be expressed in the following form:(14)$$y=F^{-1}\left(F\left(\phi \left(z\right)\cdot \phi \left(x\right)\right)⊗F\left(\alpha \right)\right)$$
where $F^{-1}$ represents the inverse Fourier transform and the maximum position of y is the location of the new frame. According to Equation (14), the KCF algorithm extracts the image block z (z is test sample in current frame) with a fixed size, which results in unsatisfactory adaptation of the original algorithm to scale changes.

The KCF is not ideal for multiscale target tracking. One can set multiple sizes and operate on each size; however, it is difficult to determine how many sizes to preset and traversing these sizes can impact the efficiency.

In the DSST tracking algorithm, two filters are proposed: a position filter and a scale filter. The scale filter can be transplanted into any other tracking algorithm without a scale change function. First of all, the optimal correlation filter h can be realized by constructing the minimum cost function using Equation (15). The input single f is d dimensional feature vector, g denotes one desired correlation output of Gaussian function, λ is the regularization parameters. In addition, $h^{l}$, $f^{l}$ and g have the same dimension and size, l∈{1,…,d}.
(15)$$\varepsilon ={\left\|\sum_{l=1}^{d}h^{l}*f^{l}-g\right\|}^{2}+\lambda {\sum_{l=1}^{d}\left\|h^{l}\right\|}^{2}$$

Equation (15) can be transformed to the Fourier domain using Parseval’s formula as shown Equation (16). The capital letters denote the discrete Fourier transform (DFT) of the corresponding vector.
(16)$$H^{l}=\frac{\overset{-}{G}F^{l}}{\sum_{k=1}^{d}\overset{-}{F^{k}}F^{k}+\lambda }=\frac{A_{t}^{l}}{B_{t}},\;l=1,\ldots ,d$$

In this paper, the scale filter in DSST is transplanted into the KCF algorithm to improve the scale change function. The DSST algorithm implements a scale search and target estimation method based on a one-dimensional independent correlation filter. In a new frame, the new candidate position of the target is determined by the classifier of KCF; then, a new scale candidate (scale feature) is obtained by using a one-dimensional scale correlation filter with the current center position as the center point. The scale selection is as follows:(17)$$\alpha^{n}P\times \alpha^{n}R,\;n\in \left\{\left[-\frac{S-1}{2}\right],\ldots \left[\frac{S-1}{2}\right]\right\}$$

In the above formula, P×R represents the size of the target in the current frame, α represents a scale factor between characteristic layers, and S represents the size of the scale filter. For each n, an image patch $J_{n}$ was extracted, the size of which is the same as that of the target that is centered at $\alpha^{n}P\times \alpha^{n}R$. The value f(n) of the training sample f is set to the d dimensional feature descriptor of $J_{n}$.

Finally, the following formula updates the scale filter $h_{scale}$ with the new sample f:(18)$$A_{t}^{l}=\left(1-\eta \right)A_{t-1}^{l}+\eta {\bar{G}}_{t}F_{t}^{l}\;B_{t}=\left(1-\eta \right)B_{t-1}+\eta \sum_{k=1}^{d}\bar{F_{t}^{k}}F_{t}^{l}$$

In this paper, the position of the candidate target is determined by the KCF tracking algorithm, and 33 scale feature samples are obtained from the target center of the current frame for training. In the new frame, the estimate of the current target state can be obtained by computing the maximum response value of the correlation filter via the inverse DFT.
(19)$$y={ℱ}^{-1}\{\frac{\sum_{l=0}^{d}\overset{-}{A^{l}}Z^{l}}{B+l}\}$$

Here, the capital letters denote that the calculation of the corresponding vector was transformed to Fourier domain, Z corresponds to the test sample z, and its feature extraction is similar to the training sample.

## 4. Tracking System Optimization

In this section, we will focus mainly on optimizing the algorithm runtime speed by utilizing limited computing resources and storage resources efficiently. The result is also applicable to other machine vision algorithms. Typically, data transmission, data parallelism and instruction parallelism directly affect the algorithm performance. Next, we conducted investigations from two aspects: data transmission and storage, and the use of data parallelism and instruction parallelism to increase the algorithm runtime speed.

### 4.1. Data Transmission and Storage

Vision DSP has a storage architecture that is composed of three layers: DDR memory (external memory), DSP local data RAM (DRAM, which belongs to the DSP cache) and a register array. The data access speed in the three-layer storage structure increases from the DDR to the register array. Table 1 presents the storage hierarchy and bandwidth. 

According to the above table, the AXI bus has the longest delay, and its bandwidth is 128 bits/cycle. From a data flow perspective, this is a bottleneck for the entire system because the local bus between the data RAM and registers provides higher throughput and lower latency. Thus, full utilization of the two AXI interfaces and saving bandwidth for reading and writing are two focuses of this paper.

As discussed in the second chapter, DSP has two 128-bit AXI interfaces: one for DSP loading/storing instructions and fetch operations and the other for iDMA to move data between the DDR and DRAM. Thus, the data storage location and transmission mode strongly affect the read/write bandwidth of the AXI interface and cycle consumption. Experiments show that the read/write bandwidth of the AXI interface is reduced by 6.4% and 11.31% via the definition of the key variables in the algorithm and the storage of the images in data RAM, respectively, compared with that in DDR (see the experiment section). This is the result of only enabling an AXI interface (in this case, iDMA does not perform a data carry operation), and the performance and bandwidth utilization of the interface are further improved if the function of iDMA is fully utilized. Next, this paper considers this perspective.

The first phase of the KCF algorithm with scale filter is mainly the training of the classifier based on the current frame, including translation filter and scale filter (the translation filter was from KCF algorithm and scale filter was from DSST algorithm). The second phase is mainly the use of the trained classifier to calculate and predict the target position and scale in the next frame. After training based on current frame, the next frame is read into the memory for prediction and classifier training. The process continues according to this strategy, as illustrated in Figure 5.

This serialization process is inefficient in data processing, especially for embedded devices that require high real-time performance. If the current-frame processing can be parallelized with the reading of the next frame, the image processing efficiency will be increased substantially. We explore a type of time-sharing strategy that is used to overcome the challenges of parallelism (DSP core processing and iDMA data transfer) and storage. In the DSP local memory, we use only one memory block to store image patch data; this memory block is called the image patch buffer. When image data are moved from the DDR to the image patch buffer, the algorithm starts to read data from the image patch buffer and extract image features. After feature processing, the image patch buffer is no longer useful for this object (after feature processing, data will be moved to other storage blocks). To fully utilize this storage, we use iDMA to transfer image data from the current frame to the image patch buffer, but this operation can only be conducted if the buffer is no longer used in the previous frame. Via this approach, the previous frame training and the data transfer operation of the current frame can be conducted in parallel, as illustrated in Figure 6.

The following is the pseudo code for the migration of the data by iDMA. First, an image frame must be read from the DDR, which can be regarded as the current frame to be processed, and iDMA must move the frame data to DRAM for processing. After entering the loop, the data of the next frame should be read into the second buffer of the ping-pong buffer to wait for migration. The pseudo code is shown in Algorithm 1:
**Algorithm 1.** Pseudo-code for migration of the data by iDMA during object trackingframenumber = 0;IndexpingPongBuffer = 0;pingPongBuffer [2];Read first frame to pingPongBuffer[IndexpingPongBuffer]; Transfer Ping buffer (pingPongBuffer [0]) to DRAM; Wait for data transmission to end;Change to Pong buffer by IndexpingPongBuffer ^ = 1;**foreach**
framenumber in frameArray[totalSize]
**do**Read next frame to Pong buffer: pingPongBuffer[IndexpingPongBuffer];if (framenumber is not first frame){Detect new object with trained template parameters.Update object position and template parameters.}Transfer Pong buffer (pingPongBuffer [1]) to DRAM;Train filter;Wait for data transmission to end;Change to Ping buffer by IndexpingPongBuffer^ = 1;**end**

DRAM is only 256 K (there are two DRAMs in a vision DSP: DRAM0 and DRAM1), and the use of DRAM requires strict limitations relative to the large capacity of DDR. How do we store a video frame in such a small space? How do we conserve the read/write bandwidth of the AXI interface? First, we must clarify that the target data we are tracking are the data we truly need. The remaining information can be regarded as redundant information. However, for the KCF tracking algorithm, due to the presence of negative samples, information around the target is still needed. Based on this, we can consider only allowing iDMA to move part of a frame for feature extraction and sample training, which conserves the AXI interface bandwidth and increases the speed of DSP image processing.

In the fifth chapter, an experiment compares the data parallelism and data serial execution. The experiment shows that the former has more advantages than the latter in terms of data transmission and processing and read/write bandwidth.

### 4.2. Using SIMD and Vector Operations to Improve the Performance

Typically, embedded hardware platforms provide invocation interfaces for software (usually called APIs) or a set of suitable operations for hardware platforms. Vision DSP also provides a corresponding API, which is usually called an intrinsic (internal function); this API shows how to make these internal calls perform well in data parallelism and instruction parallelism and improve system performance. This is another research emphasis of this paper.

The algorithm in this paper involves many complex calculations, such as sample training, video frame resizing, FFT-2D, IFFT-2D, and detection. In addition, the scale filtering algorithm is introduced in this paper; therefore, the algorithm performance in the vision DSP is especially important.

We consider an example of computing the dot product of two one-dimensional vectors to study data parallelism and instruction parallelism (refer to Algorithms 2 and 3). We compared the results before and after code vectorization and analyzed the performance changes of the code.
**Algorithm 2.** Calculate the dot product of two one-dimensional vectors of size 200void prod (short *in_a, short * in_b, short *out, short len){  int i;  for (i = 0; i++; i < len){  out[i] = in_a[i]*in_b[i];  }}

By disassembling, we observe that the above code consumes more cycles, as presented in the “Count” column in Figure 7 (approximately 1001 cycles). The large cycle count is because the function prod is not vectorized. Loop unrolling does not occur; hence, instruction execution requires more cycles in the code loop.

Next, we executed vectorization for the function prod (see Algorithm 3) and examined the cycle count via disassembly.
**Algorithm 3.** Calculate the dot product of two one-dimensional vectors of size 200. The function prod was vectorized with keyword “__restrict”void prod (short *in_a, short * in_b, short *__restrict out, short len){  int i;  #pragma aligned(in_a,64)  #pragma aligned(in_b,64)  for (i = 0; i++; i < len){  out[i] = in_a[i]*in_b[i];  }}

The results demonstrate that the cycle count of the above code is substantially lower (approximately five cycles), as presented in Figure 8.

After vectorization, the compiler will execute loop unrolling; hence, assembly instructions from the inner loop of function prod can be executed in one cycle, for instance, two load or store operations can be executed in one cycle. In this case, instruction level parallelism will be realized.

Table 2 presents the system resource consumptions before and after vectorization.

According to the statistics, the number of instructions that are issued after vectorization is 5.1% of the value prior to vectorization, and the cycle count is only 18.3% of the value prior to vectorization. Vision DSP is a multi-way SIMD processor in which each operation operates on multiple data elements in parallel and supports higher SIMD for multiply and multiply accumulate (MAC) operations; however, the precondition is that the variables must be vectorized. In the code above, the function prod without variable vectorization will be regarded as the base scalar operation. Thus, load/store operations will be executed multiple times. Consequently, the interface bandwidth between the data load/store unit and the local data memory cannot be fully used, and more cycles and the instructions must be utilized during the function execution (according to the hardware part introduction, vision DSP bundles multiple operations into an instruction bundle; this is also a precondition of instruction parallelism).

Let us consider the detection process in the KCF algorithm as an example to investigate data parallelism and instruction parallelism. Processing is simple: it uses the training sample parameters to conduct the dot product operation on the kernel space matrix (see Equation (3)). From the code implementation perspective, two level nested loops must compute the dot product, where the width and height of the image patch are both 64. Therefore, the total number of loops is 64 × 64. The time complexity of the algorithm prior to optimization is $O\left(n^{2}\right)$. To realize higher performance, the code is rewritten with intrinsics that are provided by vision DSP (see Algorithm 4). Intrinsics are C-language constructs that map machine operations, the compiler still schedules operations into VLIW instructions, and the arguments are C or vector data types.
**Algorithm 4.** Detection process of KCF tracking (calculate the response of the classifier at all shifts), and input and output parameters are vectorized through vector types and keyword “__restrict”**Input:***Kf_input_real:* The correlation coefficient real part*Kf_input_imag:* The correlation coefficient imaginary part*Alphafm_r:* The vector of sample training parameters*Size:* The product of the image patch width and height**Output:***Kf_output_real:* The correlation coefficient response real part*Kf_output_imag:* The correlation coefficient response imaginary part**Initialize:***SIMD_WIDTH = 16;*xb_vecN_2x32v* __restrict *pKf_output_real = (xb_vecN_2x32v *) ((int32_t *) Kf_output_real);**xb_vecN_2x32v* __restrict pKf_output_imag = (xb_vecN_2x32v *) ((int32_t *) Kf_output_imag);**xb_vecN_2x32v* __restrict pKf_input_real = (xb_vecN_2x32v *) ((int32_t *) Kf_input_real);**xb_vecN_2x32v* __restrict pKf_input_imag = (xb_vecN_2x32v *) ((int32_t *) Kf_input_imag);**xb_vecN_2x32v* __restrict pAlphafm_r =(xb_vecN_2x32v *) ((int32_t *) alphafm_r);*xb_vecN_2x32v *vKf_input_real, vKf_input_imag, vAlphafm_r, vKf_output_real, vKf_output_imag;*xb_vecN_2x64w *acc1;*xb_vecNx16 *temp;***Process Source-code:****foreach**
*i in Size/SIMD_WIDTH*
**do*** IVP_LVN_2X32_XP(vKf_input_real, pKf_input_real, 64);* *IVP_LVN_2X32_XP(vKf_input_imag, pKf_input_imag, 64);* *IVP_LVN_2X32_XP(vAlphafm_r, pAlphafm_r, 64);* *temp = IVP_MOVNX16_FROMN_2X32(vAlphafm_r);* *acc1 = IVP_MULUSN_2X16X32_0(temp, vKf_input_real);* *IVP_MULAHN_2X16X32_1(acc1, temp, vKf_input_real);* *vKf_output_real = IVP_PACKVRN_2X64W(acc1, 15);* *acc1 = IVP_MULUSN_2X16X32_0(temp, vKf_input_imag);* *IVP_MULAHN_2X16X32_1(acc1, temp, vKf_input_imag);* *vKf_output_imag = IVP_PACKVRN_2X64W(acc1, 15);* *IVP_SVN_2X32_XP(vKf_output_real, pKf_output_real, 64);* *IVP_SVN_2X32_XP(vKf_output_imag, pKf_output_imag, 64);***end**

Program 4 is used mainly to calculate the response values of the classifier at all shifts. The input and output parameters are vectorized through vector types and keyword “__restrict”. For example, vector type “xb_vecN_2x32v” identifies 16 32-bit signed elements, and vector type “xb_vecNx16” identifies 32 16-bit signed elements. In the loop body, the function “IVP_LVN_2X32_XP” will load aligned 16 32-bit elements (512-bit interface bandwidth) from local data memory to register files to conduct multiply operation via intrinsic functions “IVP_MULUSN_2X16X32_0” and “IVP_MULAHN_2X16X21_1”. The final response value of the classifier at all shifts will be stored by intrinsic function “IVP_SVN_2X32_XP” from narrow vector registers to local data memory and will require aligned addresses in data memory with 64-byte width. 

The optimized code has only one loop, and the time complexity is O(n). The total number of loops is 64 × 64/16, namely, the iteration count is divided by the vector length (SIMD width), and the vector registers (SIMD) are fully utilized to realize data parallelization. For example, in this case, we can load 16-way 32-bit elements directly from an aligned address, but for scalar operation, we must execute one loop 64 times. 

## 5. Conclusions Experiments and Validation Results

This chapter focuses on the experimental evaluation of the real-time performance of the tracking algorithm and the embedded system resources. In this paper, we do not attempt further improvement to the tracking algorithm; our objective is to investigate the real-time impact of vision DSP on the tracking algorithm. Hence, we must guarantee that the system-level optimization has no impact on the algorithm performance, and optimization based on vision DSP can increase the runtime speed of tracking system. So, in performance evaluation, we first evaluated simply the tracking algorithm performance with the OTB dataset(Visual Tracker Benchmark), then the real-time performance and contrast with the similar system were carried out, which are also the prerequisites of system resource evaluation.

### 5.1. Performance Evaluation

We select CarScale and Basketball sequence of the OTB dataset (OTB is the famous visual tracking benchmark, two image sequences are typical samples of deformation) to compare the effects of the scaling filter on the performance of the KCF algorithm. Figure 9 shows the poor performance when object deformation occurred. The size of the target bounding box does not change with the change of target deformation.

The problem that is caused by deformation is improved substantially via the introduction of scale filters, the size of target bounding box changes with the occurrence of target deformation, as shown in Figure 10. 

We compared the precisions of several tracking algorithms with the same video sequence: CarScale and Basketball. The statistical method of the precision is the same as that of the KCF [6]. The precision is the percentage of video frames whereby the distance between the center point of the bounding box estimated by the tracking algorithm and the center point of the ground truth is less than the given threshold. With different thresholds, the percentages obtained are different, so a curve can be obtained, that is the precision curves. The precision curves simply represent the percentages of correctly tracked frames for a range of distance thresholds. Higher precision at low thresholds corresponds to higher accuracy of the tracker. According to the test results, the precision of the KCF algorithm with the first-order gray is poor, and the KCF algorithm with the scale filter realizes a larger precision improvement, as shown in Figure 11 (the black line indicates that the KCF algorithm uses the scale estimation algorithm without using a scale filter, where the computation cost of the algorithm is high and the tracking performance is slightly worse; the blue line represents the KCF algorithm with a scale filter; the green line represents the DSST algorithm; the red line represents the KCF algorithm with first-order gray).

Table 3 presents a statistical comparison with the CarScale and Basketball sequence of the OTB dataset in terms of real-time performance. The statistics is based on X86 (CPU frequency is 2.3 GHZ) system and vision DSP (CPU frequency is 600 MHZ) separately. After applying the scaling filter to the KCF, the mean FPS(Frames Per Second) reaches 164.9192 based on vision DSP, and the mean precision is acceptable (the selected threshold is 20 pixels).

In addition, we compared several state-of-the-art embedded target tracking systems. According to Table 4, our method’s speed performance is approximately 5.5× that of the state-of-the-art embedded target tracking system.

### 5.2. System Resources

We studied the impacts of data transmission and storage on the reading and writing bandwidth of the AXI interface. Next, we calculate statistics of and analyze the system resource via three group experiments with the tracking algorithm.

(1)In the first group experiment, the key parameters in the algorithm are stored in the DDR, and every frame is read into the DDR for processing.(2)In the second group experiment, the key parameters in the algorithm are stored in the DRAM (including the image data).(3)In the third group experiment, the key parameters in the algorithm are stored in the DRAM (including the image data), and only partial frame data (image patches) are transmitted to DRAM by iDMA to conserve bandwidth and computing resources.

Prior to testing, the parameters that are used by the tracking algorithm are configured as follows: λ is equal to 1 × 10^−4^, the interpolation factor is equal to 0.075, the Gaussian kernel bandwidth is equal to 0.2, and padding is equal to 1.5. For the experiment, frame sequences of a basketball game are selected from the OTB dataset.

The experimental results demonstrate that the average numbers of cycles are 20,310,278.98 for the first group of experimental data, 7,086,780.375 for the second group, and 2,331,262.1 for the third group. Figure 12 clearly shows that the third group consumed the fewest cycles (only 11.5% as many as the first group and 32.9% as many as the second group).

For the third group of experiments, to conserve the data read/write bandwidth and increase the data processing efficiency, only the image patch (106 × 222) of every frame was moved by iDMA to DRAM, whereas for the other two groups of experiments, no such operation was conducted. The final read/write bandwidth and performance are presented in Table 5.

For the second group experiment, the KCF algorithm parameters were saved in DRAM (typically, the data use frequently is saved in DRAM to increase the data processing speed). Thus, every frame will be swapped to DRAM, but in the first group experiment, the parameters will be swapped between DRAM and DDR in addition to every frame, which will cause the AXI interface for DSP loading and storage instructions to be occupied. Thus, in the first two group experiments, the bandwidth has higher loads. In contrast, in the third group experiment, the data will be swapped by iDMA; hence, only another AXI interface, which is mainly used to move data, is occupied, which causes less bandwidth to be occupied for the interface of loading and storage instructions.

In addition, according to the statistics in Table 5, the frame rate will be determined by two factors: the data storage and the data transmission. In the second experiment, the parameters that are used frequently are allocated in DRAM, and the frame rate is substantially higher than that of the first group. In the third group experiment, the interface of data transmission is utilized fully, and the parallelization of data transmission and processing directly increases the frame rate.

## 6. Conclusions

The strongest advantage of vision DSP is used for a computer vision application, but it is difficult to fully analyze the advantages of this processor by using only a common programming approach. Therefore, this paper has conducted basic research in this field.

First, if the data are processed via the typical serial approach, inefficient data processing occurs, and the data reading and writing of the AXI interface also occupy a large bandwidth. This paper has conducted basic research on this and has compared the serial data transmission and parallel transmission via iDMA. The experimental results demonstrated that the latter far outperforms the former.

Second, data parallelism and instruction parallelism are the key factors for improving the performance of tracking algorithms. Examples are presented to demonstrate the implementation of vectorization at the code level and to analyze the corresponding instructions to study data parallelism and instruction parallelism.

Finally, the data bandwidth from DDR to the DSP local data memory (DRAM) is examined in this paper, which is significant for conserving bandwidth resources and system computing resources. In addition, we have applied a scale filter to the KCF tracking algorithm to improve its performance in scale transformation. The experimental results demonstrate that the improved algorithm can cope with scene scale transformation.

Two research directions are identified for future investigation:(1)We investigated the impact factors of an embedded system for single target tracking with limited resources. For multi-object tracking (MOT) [29,30,31,32], we must consider the efficient utilization of the limited DSP local data memory, as more tracking objects will require more memory.(2)Visual DSP allows applications to use internal instruction sequences indirectly through APIs, and DSP also allows users to customize instruction sequences. The identification of optimal instruction sequences is a valuable research objective.

## Figures and Tables

**Figure 1 sensors-20-06494-f001:**
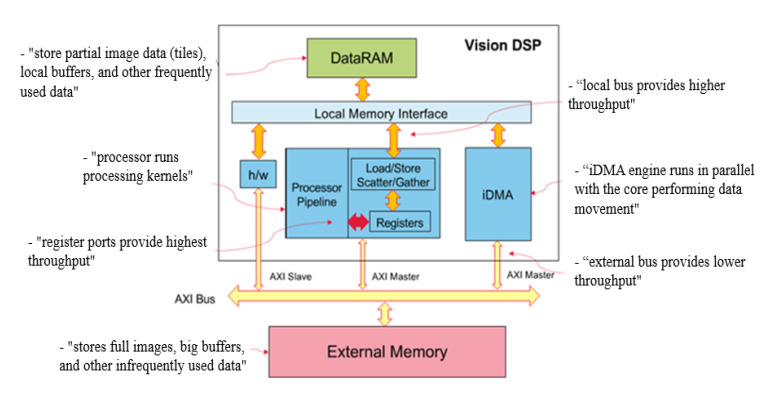
Vision digital signal processor (DSP) Block Diagram (the illustration is from Cadence Company technical material).

**Figure 2 sensors-20-06494-f002:**
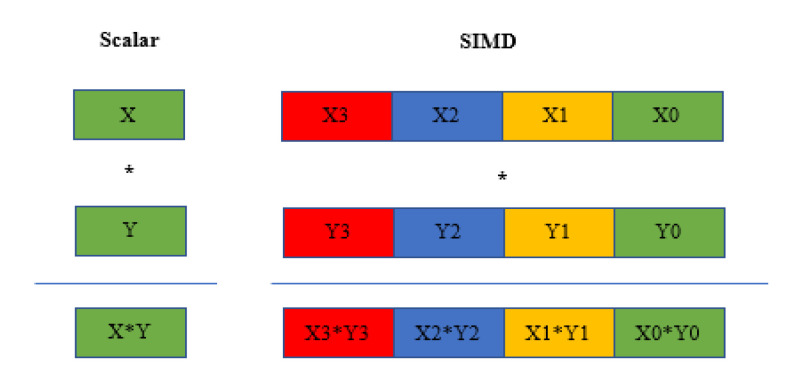
Scalar and single instruction multiple data (SIMD) multiplication comparison (the illustration is from Cadence Company technical material).

**Figure 3 sensors-20-06494-f003:**
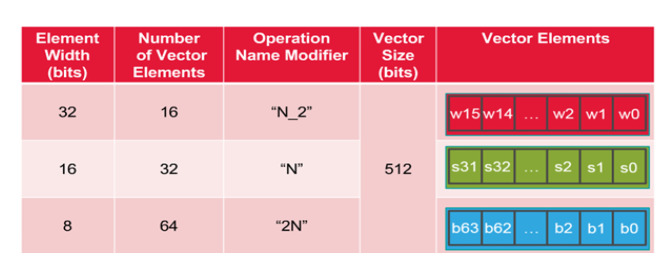
Vision DSP vector register (the illustration is from the Cadence Company technical material). “N_2” corresponds to 16 32-bit signed elements, “N” to 32 16-bit signed elements, and “2N” to 64 8-bit elements.

**Figure 4 sensors-20-06494-f004:**
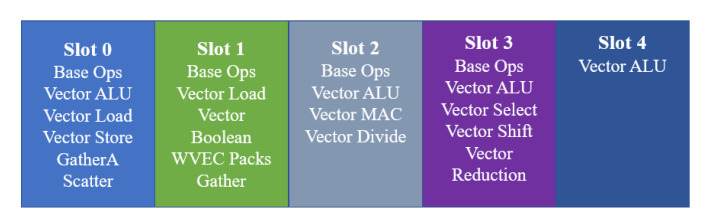
Five instruction issue slots and operations of vision DSP (the illustration is from the Cadence Company technical material).

**Figure 5 sensors-20-06494-f005:**
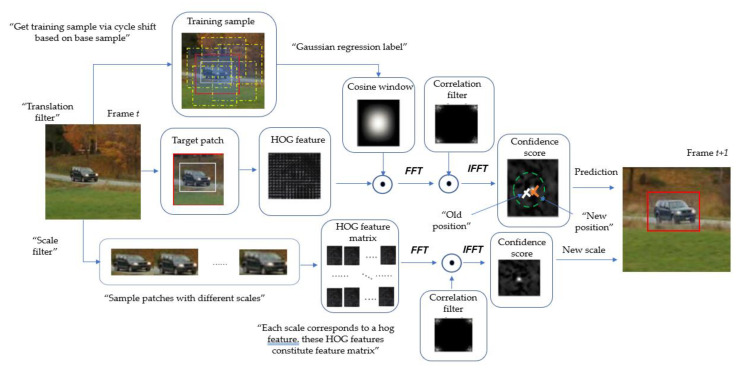
Training of classifier and target prediction process. For translation filter, training sample was generated via cycle shift (yellow dotted box) based on base sample and training the filter with target histogram of oriented gradients (HOG) or gray feature. For scale filter, samples of different scales are constructed based on the target position, and these samples will be used as the training samples of the scale filter. “*FFT*” and “*IFFT*” denote fast Fourier transform and inverse fast Fourier transform, respectively.

**Figure 6 sensors-20-06494-f006:**
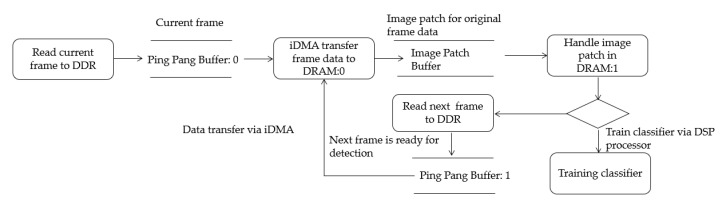
Processing sequence of frame data.

**Figure 7 sensors-20-06494-f007:**
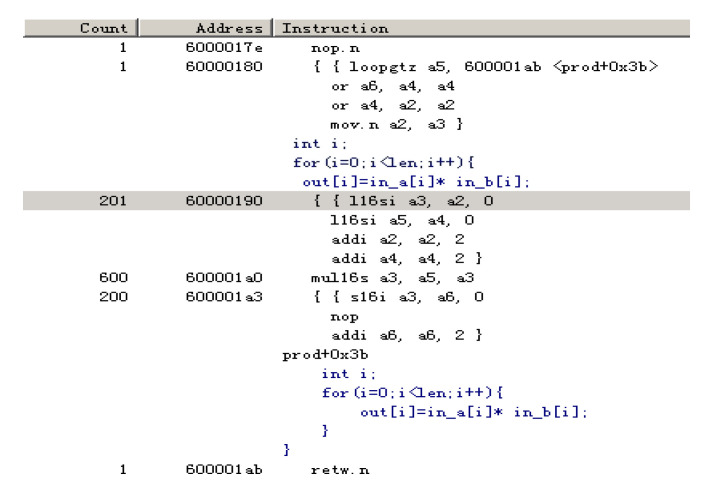
Cycle count statistics of dot product of two one-dimensional vectors without vectorization.

**Figure 8 sensors-20-06494-f008:**
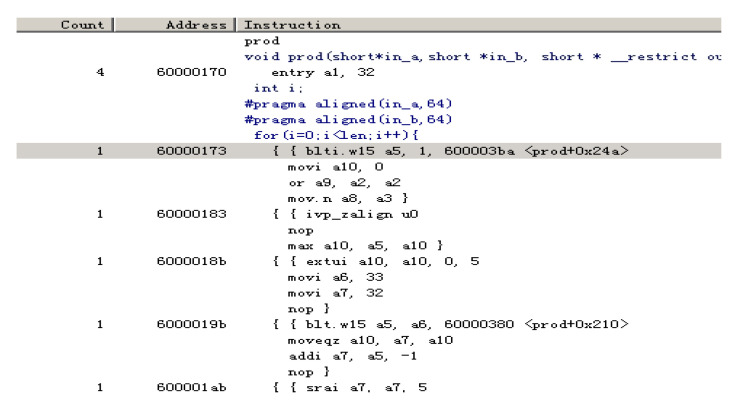
Cycle count statistics of dot product of two one-dimensional vectors with vectorization.

**Figure 9 sensors-20-06494-f009:**
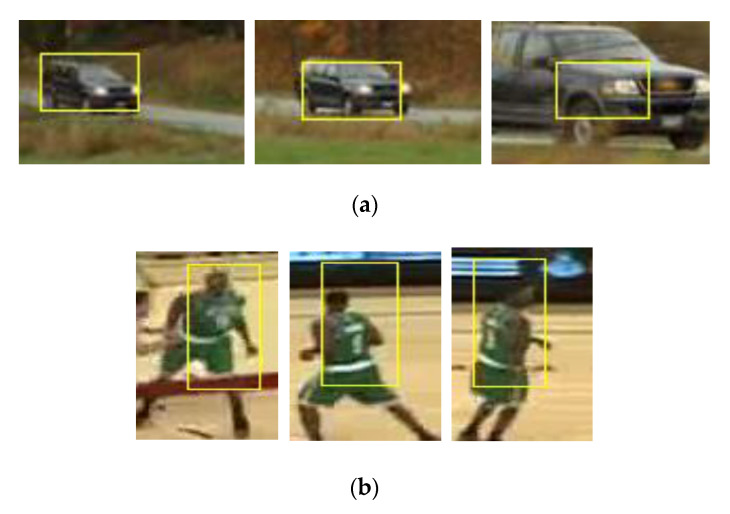
KCF without the scale filter. From the left to the right, (**a**) CarScale sequence: the results are sampled at frame 12, 57 and 176, respectively; (**b**) Basketball sequence: the results are sampled at frame 5, 137 and 204, respectively.

**Figure 10 sensors-20-06494-f010:**
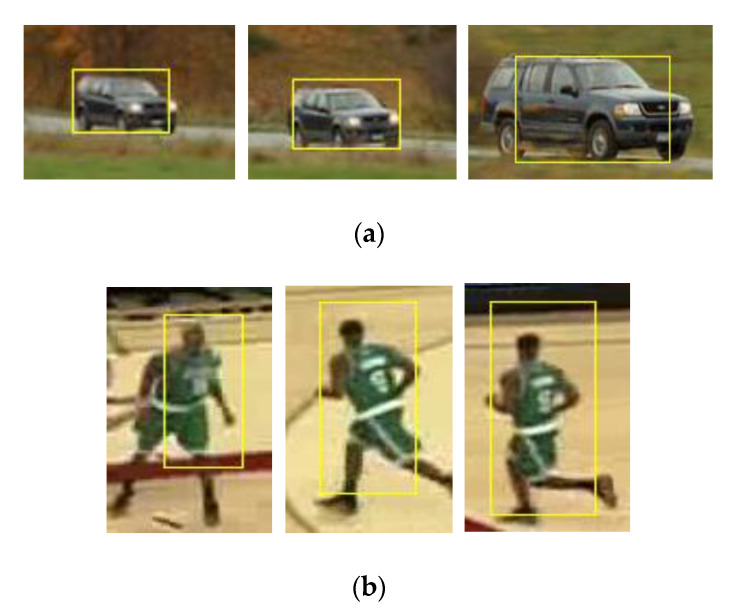
Introduction of the scale filter into the KCF. (**a**) CarScale sequence: the results are sampled at frame 12, 57 and 180, respectively; (**b**) Basketball sequence: the results are sampled at frame 5, 107 and 132, respectively.

**Figure 11 sensors-20-06494-f011:**
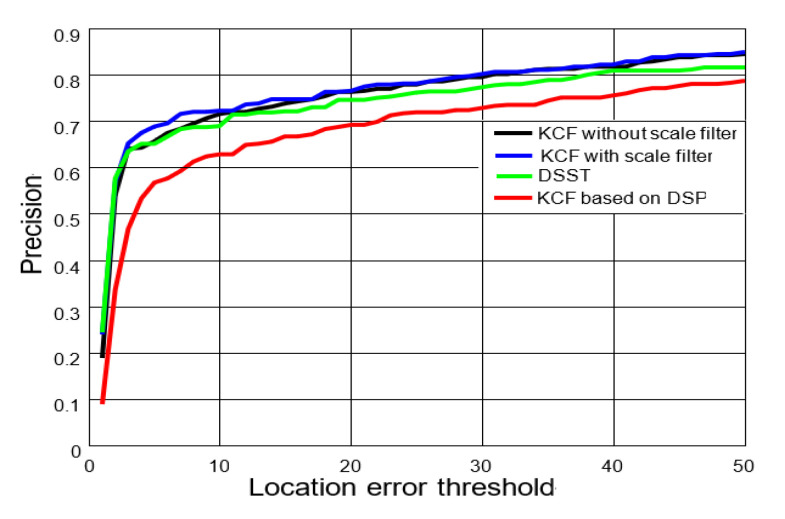
Precision comparison. The *x*-axis and *y*-axis denote the distance threshold in pixels and precision (percentages of correctly tracked frames for a range of distance thresholds), respectively.

**Figure 12 sensors-20-06494-f012:**
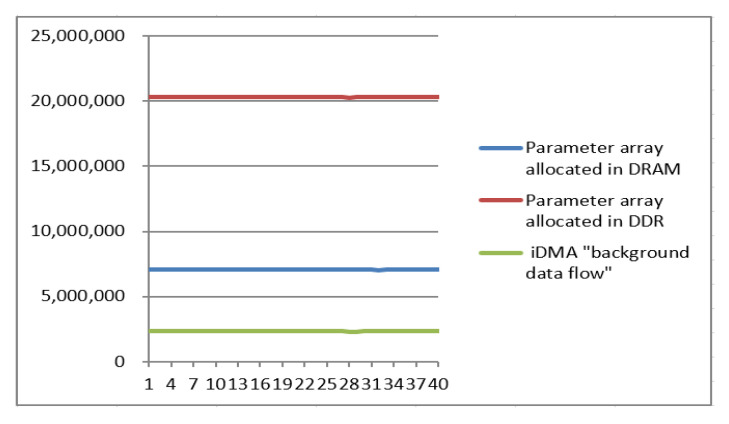
Cycle statistics, whereby the horizontal coordinate represents the video sequence and the vertical coordinate represents the number of cycles that were consumed.

**Table 1 sensors-20-06494-t001:** Vision DSP memory hierarchy.

Memory	Size	Connection	Bandwidth	Latency
Register	~3.5 KB	Register Port	>1 KB/Cycle	Direct access
Data RAM	32 KB~1 MB	Local Bus	128 B/Cycle	Low
External memory	4 GB	AXI Bus	16 B/Cycle	High

**Table 2 sensors-20-06494-t002:** Consumption of system resources by the dot product operation.

Function	Vectorizable?	Load	Store	Cycle	Instructions
Prod	N	449	224	1001	600
Prod	Y	65	31	5	31

“Vectorizable” indicates whether the variables that are involved in the function are vectored, “Cycles” represents the number of cycles that are consumed by the function, the “Load” and “Store” columns specify the numbers of load and store operations, respectively, and “Instructions” indicates the number of instructions that are required for this function.

**Table 3 sensors-20-06494-t003:** Performance comparison.

Algorithm	Feature	Mean Precision (20 px)	Mean FPS	CPU Frequency
KCF	HOG + Scale filter	0.7539	88.6548	2.3 GHZ
DSST	HOG + Scale filter	0.7550	89.2422	2.3 GHZ
KCF	HOG + Scale filter	0.7546	164.8192	600 MHZ
KCF	HOG + Scale estimation	0.7419	171.5030	2.3 GHZ

**Table 4 sensors-20-06494-t004:** Speed performance comparison.

Algorithm	Hardware	Optimization Method	Mean FPS	CPU Frequency
KCF [22]	FPGA+ARM	Parallel calculation with multi-core heterogeneous processors	30 fps	1.2 GHZ
Object tracking [23]	CPU+FPGA	Optimizes the performances of the background subtraction by GMM(Gaussian Mixture Model)	15.9 fps (640 × 480)	667 MHZ
KCF [24]	TMS320C6414	Hardware adaptability optimization	25 fps	850 MHZ
Improved KCF [26]	TMS320C6678	Multi-core parallel	30 fps	1.25 GHZ
Object tracking [27]	FPGA	Fusion algorithm of background difference and Sobel	60 fps (640 × 480)	N/A
Object tracking [28]	FPGA	Binarized Depthwise Separable Neural Network + Hardware optimization	11.1 fps	N/A
Ours	Vision DSP	Hardware optimization + Storage and transmission policy	164.9 fps	600 MHZ

**Table 5 sensors-20-06494-t005:** Bandwidth and performance statistics.

Input Image (576 × 432)	Read (bandwidth/frame)	Write (bandwidth/frame)	Total (bandwidth/30 fps)	DSP Performance (600 MHZ)
First Group	45.29	21.83	2013.6 MB/s	29.56 fps
Second Group	42.37	19.36	1851.9 MB/s	84.75 fps
Third Group	1.63	1.44	92.1 MB/s	164.9 fps

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
