# Peer review of "Study on the Moving Target Tracking Based on Vision DSP"

_sensors, 2020, doi:10.3390/s20226494_

Round 1
Reviewer 1 Report
This paper embeds KCF tracker into vision DSP. The following issues should be addressed.
- Please clearly clarify the contributions of this paper.
- Please conduct extensive evaluations on benchmark datasets, such as OTB100, VOT2018, etc.
- Can the method readily extend to other tracking algorithms, such as Parallel attentive correlation tracking, Robust visual tracking via convolutional networks without training?
- What are the limitations of the proposed method ?
Reviewer 2 Report
The work is interesting.
I would, however, discuss and cite similar works from 2020 and would decrease the references in Chinese as the journal is international and not all people know Chinese language.
The algorithms and computer programs are presented very well, but I would prefer seeing their computer flowcharts
Reviewer 3 Report
General Comments:
The paper handles visual tracking systems and presents a study on evaluating hardware optimization methods that provide efficient utilization under limited hardware resources. This paper also studies the performance of kernel correlation filter tracking algorithm based on a vision digital signal processor. The topic is important for moving target tracking systems and the presented study is useful. However, clarity and lack of referencing are the main issues in this paper. A major revision is necessary to clarify the presented methods and specify the proper references.
Specific Comments:
- Section 3.1:
- All formulas in this section should be referenced and explained clearly. The same comment applies to Section 3.2.
- Better explanation is required to show how tracking is performed; also, how the correlation coefficient is calculated.
- Section 4.1:
- Training of the classifier is unclear.
- Image features should be defined and clarified.
- Section 5:
- A clear performance measure is required.
- In Figure 10, the term “precision” should be clearly defined.
Round 2
Reviewer 3 Report
The Authors have performed a major revision on this manuscript, where they addressed the Reviewers’ comments in detail. The current version of the manuscript is useful and ready for publication.